REGISTERED REPORT PROTOCOL

# Protocol for the systematic review of the epidemiology of superficial Streptococcal A infections (skin and throat) in Australia

**Sophie Wiegele**[1]*, **Elizabeth McKinnon**[2], **Rosemary Wyber**[2], **Katharine Noonan**[2]

**1** Clinical Medicine, Perth Children's Hospital, Perth, Western Australia, Australia, **2** Research, Telethon Kid's Institute, Perth, Western Australia, Australia

* Sophie.Wiegele@health.wa.gov.au

## Abstract

### Objective

We have produced a protocol for the comprehensive systematic review of the current literature around superficial group A Streptococcal infections in Australia.

### Methods

MEDLINE, Scopus, EMBASE, Web of Science, Global Health, Cochrane, CINAHL databases and the gray literature will be methodically and thoroughly searched for studies relating to the epidemiology of superficial group A Streptococcal infections between the years 1970 and 2019. Data will be extracted to present in the follow up systematic review.

### Conclusion

A rigorous and well-organised search of the current literature will be performed to determine the current and evolving epidemiology of superficial group A Streptococcal infections in Australia.

## Introduction

*Streptococcus pyogenes* (Group A beta-haemolytic streptococci, GAS) is a human only pathogen which causes a broad range of superficial, invasive and post-infectious conditions. The most common infections are superficial (sGAS), affecting the skin or throat [1, 2].

Skin infection, variously described as impetigo, pyoderma or skin sores, is a common disease of childhood. The median prevalence globally is 12.3%, with a higher prevalence (13–90%) documented in remote Australian Aboriginal and Torres Strait Islander communities [3–11]. Skin sGAS infections are characterized by honey-coloured crusting lesions of the skin and present as either primary disease or as a secondary infection of an underlying skin condition [12]. Impetigo can manifest as single or multiple non-healing ulcers on any part of the body, often precipitated by a disruption to skin integrity [12, 13]. *Staphylococcus aureus* is the

**Data Availability Statement:** All relevant data from this study will be made available upon study completion.

**Funding:** SW $1300 USD (SW) Telethon Kid's Institute, Western Australia https://www.telethonkids.org.au/ The funders had and will not have a role in study design, data collection and analysis, decision to publish, or preparation of the manuscript.

**Competing interests:** The authors have declared that no competing interests exist.

most common causative organism in temperate regions, compared with tropical regions where *Streptococcus pyogenes*, a group A beta-haemolytic streptococcus, predominates [2, 12, 14]. Current evidence suggests that scabies infestation, tropical climate and crowding increases the prevalence of pyoderma, and that factors including treatment of scabies infestation, good skin health and swimming pool exposure may be protective for impetigo [3, 14–23].

Similarly, throat infection, including pharyngitis and tonsillitis are common infections affecting children and young adults and account for approximately 3% of presentations to general practitioners in Australia [24, 25]. There is limited existing data about these presentations in remote Aboriginal and Torres Strait Islander settings. Pharyngitis is characterised predominately by sore throat and difficulty swallowing. Other symptoms may include tender cervical lymph nodes and fever. On examination an erythematous pharynx may be found or enlarged and erythematous tonsils indicating tonsillitis. Pharyngitis is caused by a range of pathogens, most commonly respiratory viruses, but GAS can be isolated in up to 20% of symptomatic children and 10% of asymptomatic children [26, 27].

sGAS infections disproportionately affect Indigenous, geographically remote and marginalised populations globally [3, 5, 16, 28]. These disparities have been attributed to risk factors including low socioeconomic status, overcrowded housing, resource limited settings and inadequate access to primary health care [3, 28].

sGAS infections have a direct impact on health causing pain and likely reducing education participation. Skin sores are also associated with stigma in some Aboriginal and Torres Strait Islander communities [29]. sGAS infections have potential to progress to more significant local disease, with skin infections leading to cellulitis and throat infection to peri-tonsillar abscess. More serious sequelae may also develop including invasive disease (sepsis, necrotising fasciitis, osteomyelitis, septic arthritis) and toxin mediated response (Streptococcal Toxic Shock Syndrome, Scarlet Fever), which currently account for a significant burden of disease in Australia (5 in 100 000 children annually) [1, 30]. sGAS infections also have the potential to precipitate abnormal an immune response resulting in acute post-streptococcal glomerulonephritis (ASPGN) and acute rheumatic fever (ARF) [14, 31–36].

The consequence of single or multiple episodes of post-streptococcal autoimmune sequelae may be the development of chronic disease: Chronic Kidney Disease (CKD) in ASPGN and Rheumatic Heart Disease (RHD) in ARF [37–39]. ASPGN is a significant cause of CKD, and has an incidence of 13.6–17.0/100 000 person-years in the Northern Territory, increasing to 124.0/100 000 person-years in Aboriginal children [40–42]. RHD prevalence in Australia is 845.2/100 000 people vs 5.2/100 000 people in Indigenous vs non-Indigenous populations, showing a clear disparity in the rate of preventative chronic heart disease in these populations [37]. There is significant morbidity and mortality associated with RHD, particularly for people under the age of 50, as well as the economic burden of ongoing treatment and surgery [37].

The disproportionate disease burden of sGAS infections among Aboriginal and Torres Strait Islander communities in Australia has spurred a range of research initiatives over many years. A smaller number of studies have also addressed sGAS infections in non-Indigenous settings in Australia. Although there is a relatively large body of research from this work no synthesis of these studies has been conducted to provide national and subnational estimates of the burden of disease. This makes it difficult to assess sGAS changes over time and stymies disease control efforts. Therefore, this systematic review aims primarily to assess the incidence and prevalence of sGAS infections in Australia. As secondary outcomes, the incidence and prevalence of sGAS infections in focus populations within Australia will be reviewed including Aboriginal and Torres Strait Islander communities, rural and remote locations, age group and climate type.

## Methods

### Search strategy and selection criteria

This systematic review is reported according to the Preferred Reporting Items for Systematic Reviews and Meta-analyses (PRISMA) protocols statement and the Meta-analysis of Observational Studies in Epidemiology (MOOSE) guideline [43, 44]. No protocol for this systematic review currently exists.

Articles for data extraction will be identified through a search of multiple sources. A systematic search for published studies will be performed by SW, a medical doctor with assistance from researchers and librarians, through MEDLINE, Scopus, EMBASE, Web of Science, Global Health, Cochrane and CINAHL. A grey literature search will be done through Google Scholar top 1000 articles sorted by most relevant, WHO IRIS library database, Trove, Research Data Australia, the Grey Literature Report, and Australian Infection Prevention and Control. A search will also be completed of clinical trial registries including the Cochrane Central Register of Clinical Trials, WHO International Clinical Trials Registry Platform, Australian New Zealand Clinical Trial Registry and ClinicalTrial.gov. A hand search of included articles and related systematic reviews will be performed. All searches will be restricted to papers between the years of 1970 and the end of 2020. A detailed search strategy of electronic databases can be reviewed in S1 Appendix.

The inclusion criteria are: population based study, incidence and/or prevalence data available, Australian population, objective and transferable data on people with: impetigo, pyoderma, pharyngitis, tonsillitis, superficial GAS infection, non-specific sore throat, and a term including numerator and population denominator or a specific summary of incidence and prevalence used to describe this cohort. Studies that were pathogen-specific, other than Streptococcal A, where participants numbered less than 20 and those describing a cohort not including children (defined as 0–16 years) will be excluded.

### Data extraction

Data will be extracted by two independent reviewers (SW, SE or BVS) and extracted data entered into a Google Form. Data points will be compared and discrepancies resolved between data extractors or a third party reviewer. Variables for extraction will be subdivided into groups: article details/ timing, location, study design, demographics, environmental/ social factors, skin infection specific variables and throat infection specific variables.

Article details/ timing encompasses data points including title, authors and duration of study. Location information will include state, city and community levels. Data on Australian Statistical Geography Standard (ASGC) region will be entered post-extraction. This system uses data from the Australian Bureau of Statistics (ABS) to classify remoteness based on a combination of population size and access to services, dividing communities into 5 classes: Major City, Inner Regional, Outer Regional, Remote and Very Remote.

Variables in the study design subset of extraction include: type of study, sampling method, qualifications of person collecting data, population type and total number of participants. The demographics of the study population will be collected through data points of population age, number of males and number of Aboriginal and Torres Strait Islander people in the population.

Specific data points for extraction from environmental and social factors include description of possible positive and negative factors contributing to the epidemiology of superficial Streptococcal A infections. Post extraction data entry for climate will be performed using the Koppen Climate Classification System, dividing climate into tropical, temperate, arid or cold [45, 46].

Data on the study's definition of skin infection will be recorded, as well as epidemiological data including individual infection rate and base population number, presentation rates, total number of presentations and frequency with age. Data specific to impetigo will also be collected including distribution, number and type of lesions, as well as whether a precipitating lesion, concurrent scabies or throat infection were identified. In a similar way, data points on the definition of throat infection and epidemiological data will be extracted.

A full list of data points and any accompanying definitions or simplifications can be found in S2 Appendix.

## Outcomes

The main outcome of this systematic review is to define the burden of disease of s-GAS infections in Australia, which primarily include incidence and prevalence data.

Secondary outcomes from the paper include factor specific epidemiological data; defining the burden of disease in sub-populations which include: climate, remote status, Aboriginal and Torres Strait Islander background, presence of social and/or environmental factors and year of study.

## Assessment of bias

Bias will be assessed for each study included in the systematic review by two independent reviewers (SW, BVS). The Joanna Briggs Institute critical appraisal checklist for prevalence studies will be applied to each included study to assess for coverage, measurement and classification bias [47]. Any additional statement of bias from the study will be documented. Sensitivity analysis will consider overall quality of each study as well as the definitions for disease. Initial analysis will include all articles.

## Statistical analysis and synthesis

Prevalence and incidence data will be extracted and presented for the skin and throat cohorts. Subgroup analysis will explore variation in incidence and prevalence data according to: geography (urban, remote), Aboriginal and Torres Strait Islander background, climate type, age of population, year of study. Within each cohort (skin and throat) descriptive analysis will include environmental/ social contributors to disease, disease characteristics and group A streptococcal isolation.

Pooled prevalence estimates across studies in a similar setting (community-based or health service encounters) will be generated using a random effects model and heterogeneity between studies will be evaluated by means of a $\chi^2$ test on Cochran's Q statistic and quantified with the $I^2$ statistic. A random effects framework is deemed appropriate given studies in this field sample from different populations which have varying characteristics. Metaregression will be undertaken to assess sources of between-study heterogeneity in the pooled prevalence estimates. Publication bias will be assessed with the aid of funnel plots and Begg's rank correlation test.

## Assessment of meta-biases

Bias will be assessed according to the Cochrane handbook [48]. Firstly, a rigorous grey literature and hand search will reduce the risk of publication bias. Time lag bias will be considered, though as this paper is focused around a long-standing disease in Australia and will include a publication range of 49 years, we have assessed the risk of this bias to be small. In an effort to reduce duplication bias, the dataset of each included paper will be carefully scrutinised to

detect multiple publications using the same dataset. Obvious duplications in datasets will result in the exclusion of the later paper. In the event of any possible duplications not clearly outlined by the paper, the authors of the papers will be contacted.

Location bias will be addressed by instituting a comprehensive search strategy of seven main-stream databases and six grey literature databases as can be seen in S1 Appendix. The search will be conducted for all languages and journals, with papers not available online obtained in hard-copy format. Although all papers are based in Australia, there may be regional differences in publication rates based on states and territories leading to region specific location bias. The geographical distribution of selected papers will be clearly defined, and detail areas of dense and sparse literature coverage that may contribute to bias. Rigorous bibliographic review will be undertaken of relevant systematic reviews and included articles, when combined with our com-prehensive search strategy will not increase the risk of citation bias.

As this study focuses on Australian disease epidemiology, Australia being a predominantly English speaking country, language bias is likely to be low. Regardless of this, the search strat-egy will not be limited to English studies, reducing the risk of language bias. As this review focuses on papers with descriptive statistics, and the inclusion criteria are results based, papers will be included irrespective of primary outcomes.

## Confidence in cumulative evidence

Each paper will be assessed for confidence by two independent reviewers (SW, BVS) using the Joanna Briggs Institute critical appraisal checklist [49]. A final assessment of confidence in the paper will be assessed by the two independent reviewers, assigning quality levels: very poor, poor, fair, fair to good and good, to each paper. The levels of confidence will be reported for each paper included in the systematic review.

## Conclusion

A rigorous and well-organised search of the literature will be performed to determine the cur-rent evidence on the epidemiology of s-GAS infections in Australia. As outlined above, all efforts will be taken to reduce bias and produce a collection of papers for a reliable and valid meta-analysis of the current data. Ultimately, the results of this paper will guide the allocation of health resources to areas where the development of preventative measures will most greatly reduce the burden of s-GAS infections and post-Streptococcal sequelae.

## Supporting information

**S1 Appendix. Search strategy.**
(DOCX)

**S2 Appendix. Data extraction points and definitions.**
(DOCX)

**S3 Appendix. Joanna Briggs Institute critical appraisal checklist for studies reporting prev-alence data.**
(DOCX)

**S1 File.**
(PDF)

**S1 Checklist.**
(PDF)

## Acknowledgments

The authors would like to acknowledge and thank Ms Stephanie Enkel, Dr Asha Bowen, Mr Jeffrey Cannon (Telethon Kids Institute) and Mr Bede van Schaijik (University of Western Australia) for their assistance in the design of the search strategy for this protocol. Advisory support from the Telethon Kids Institute was provided in the development of this protocol. No funding was received for the development of this protocol.

## Author Contributions

**Conceptualization:** Sophie Wiegele, Elizabeth McKinnon, Rosemary Wyber.

**Methodology:** Sophie Wiegele, Rosemary Wyber.

**Project administration:** Sophie Wiegele, Rosemary Wyber.

**Resources:** Elizabeth McKinnon.

**Software:** Sophie Wiegele.

**Supervision:** Rosemary Wyber.

**Visualization:** Sophie Wiegele.

**Writing – original draft:** Sophie Wiegele, Katharine Noonan.

**Writing – review & editing:** Sophie Wiegele, Elizabeth McKinnon, Rosemary Wyber, Katharine Noonan.

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
