## [Decision Letter · Decision Letter 0]

9 Mar 2021

PONE-D-20-37209

Protocol for the systematic review of the epidemiology of superficial Streptococcal A infections (skin and throat) in Australia.

PLOS ONE

Dear Dr. Wiegele,

Thank you for submitting your manuscript to PLOS ONE. After careful consideration, we feel that it has merit but does not fully meet PLOS ONE’s publication criteria as it currently stands. Therefore, we invite you to submit a revised version of the manuscript that addresses the points raised during the review process.

Please follow reviewers comments one by one. In addition see below:

- Report also according to MOOSE guidelines

- Exclusion criteria – consider excluding studies reporting less than 20 participants

- To support your choice of the Joanna Briggs Institute critical appraisal checklist for risk of bias assessment (see reviewer comments) you can cite: Migliavaca et al. Prevalence Estimates Reviews – Systematic Review Methodology Group (PERSyst). Quality assessment of prevalence studies: a systematic review. J Clin Epidemiol. 2020 Nov;127:59-68.

- I agree with the reviewers' comments on "If bias is assessed to have influenced the outcomes described in the paper, it will be excluded." -  all relevant studies fulfilling inclusion criteria should be included and a sensitivity analysis according to risk of bias assessment should be performed.

We look forward to receiving your revised manuscript.

Kind regards,

Dafna Yahav

Academic Editor

PLOS ONE

Journal Requirements:

2) We note that you have stated that you will provide repository information for your data at acceptance. Should your manuscript be accepted for publication, we will hold it until you provide the relevant accession numbers or DOIs necessary to access your data. If you wish to make changes to your Data Availability statement, please describe these changes in your cover letter and we will update your Data Availability statement to reflect the information you provide.

3) Please include captions for your Supporting Information files at the end of your manuscript, and update any in-text citations to match accordingly. Please see our Supporting Information guidelines for more information: http://journals.plos.org/plosone/s/supporting-information.

4)  PLOS considers systematic reviews and related protocols that will provide an up-to-date analysis of a particular research question. We typically define "up-to-date" as within the last 12 months. Therefore, please revise your methods that state that publications up until 2019 will be searched.

5)  Please note that appendices are in the body of the manuscript (at the end), and it might be more appropriate to include these as supplementary files. Please evaluate.

Reviewers' comments:

Reviewer's Responses to Questions

**Comments to the Author**

1. Does the manuscript provide a valid rationale for the proposed study, with clearly identified and justified research questions?

Reviewer #1: Yes

Reviewer #2: Yes

2. Is the protocol technically sound and planned in a manner that will lead to a meaningful outcome and allow testing the stated hypotheses?

Reviewer #1: Yes

Reviewer #2: Yes

3. Is the methodology feasible and described in sufficient detail to allow the work to be replicable?

Reviewer #1: Yes

Reviewer #2: Yes

4. Have the authors described where all data underlying the findings will be made available when the study is complete?

Reviewer #1: Yes

Reviewer #2: Yes

5. Is the manuscript presented in an intelligible fashion and written in standard English?

Reviewer #1: Yes

Reviewer #2: Yes

6. Review Comments to the Author

You may also provide optional suggestions and comments to authors that they might find helpful in planning their study.

Reviewer #1: This article describes a protocol for conducting a systematic review of the epidemiology of superficial streptococcal A infections in Australia. The article is well written and meets the requirements of the PRISMA statement for the protocols of a systematic review. On the other hand, in the introduction, the authors adequately and extensively justify the need and convenience of carrying out this systematic review. They include adequate search strategies in multiple databases. Therefore, I think it deserves to be published. A few minor points need to be clarified:

-In line 103, there seems to be a repetition of the ideas in two consecutive sentences. They could try to avoid these repetitions.

-in line 134, it is stated that there is no risk of country-based location bias. Despite this, Australia is a large country, and it would be necessary to discuss whether there is a risk of bias based on the different territories of the country.

Reviewer #2: Thanks for the opportunity to review this protocol. protocol for a systematic review for Streptococcal infections. The protocol is well done and my comments are minimal.

1. In assessing study quality, why not consider measures that systematically evaluate bias in the design and implementation of the studies? Such as the Cochran’s risk of bias Tool?

2. Is it standard practice to exclude studies on the basis of bias? Why not exclude the, only in sensitivity analysis?

3. The terms time lag bias and duplication bias are unfamiliar to me. Please provide references for their use.

4. Provide justification for why you’re not considering fixed effects models.

5. Provide citation for the climate classification system and a brief justification for your choice.

6. Will study selection be done in duplicate with a system for breaking ties?

7. The introduction section is lengthy and can be summarized.

8. Study years – please update to end of 2020 at the latest.

7. PLOS authors have the option to publish the peer review history of their article (what does this mean?). If published, this will include your full peer review and any attached files.

Reviewer #1: No

Reviewer #2: No

---

## [Author Response · Author response to Decision Letter 0]

14 Jul 2021

Firstly, thank you for your input and feedback on this paper. The suggestions have been taken on board and can be viewed in the ‘Revised Manuscript with Track Changes’ document. 

In response to the specific reviewer comments:

Reviewer 1

In line 103, there seems to be a repetition of the ideas in two consecutive sentences. They could try to avoid these repetitions.

A portion of this statement has been deleted to better summaries the point and the introduction.

In line 134, it is stated that there is no risk of country-based location bias. Despite this, Australia is a large country, and it would be necessary to discuss whether there is a risk of bias based on the different territories of the country

The country-based location bias statement has been updated. It now states that “Although all papers are based in Australia, there may be regional differences in publication rates based on states and territories leading to region specific location bias. The geographical distribution of selected papers will be clearly defined, and detail areas of dense and sparse literature coverage that may contribute to bias.” 

Reviewer 2

In assessing study quality, why not consider measures that systematically evaluate bias in the design and implementation of the studies? Such as the Cochrane’s risk of bias Tool?

For this systematic review we have chosen to use the Joanna Briggs Institute critical appraisal checklist for risk of bias assessment. The full checklist is documented in the Appendix 3 of the protocol paper and systematically covers multiple areas of bias. 

The justification of this choice can be supported by the systematic review of the quality assessment of prevalence studies by Migliavaca et al. which states “among the currently available tools specific for prevalence studies, the Joanna Briggs Institute Prevalence

Critical Appraisal Tool has a higher methodologic rigor and addresses what we consider the most important items related to the methodological quality of prevalence studies and may be considered the most appropriate tool”. We have changed the paper to include this reference in support of the choice.

Is it standard practice to exclude studies on the basis of bias? Why not exclude the, only in sensitivity analysis?

Sensitivity analysis will consider overall quality of study as well as definitions for disease. Initial analysis will include all articles.

The terms time lag bias and duplication bias are unfamiliar to me. Please provide references for their use.

Both are according to the Cochrane Handbook for Systematic Reviews of Interventions.

Time-lag bias: “The rapid or delayed publication of research findings, depending on the nature and direction of the results.”

Duplication bias: “The multiple or singular publication of research findings, depending on the nature and direction of the results.” 

Isabelle Boutron, Matthew J Page, Julian PT Higgins, Douglas G Altman, Andreas Lundh, Asbjørn Hróbjartsson; on behalf of the Cochrane Bias Methods Group. Cochrane handbook for systematic reviews of interventions, Chapter 7: Considering bias and conflicts of interest among the included studies. Cochrane Training. 2021; Part 2 (7).

Provide justification for why you’re not considering fixed effects models.

A random effects model is deemed appropriate given studies in this field sample from diverse populations, even though we are restricting to Australian studies.

Provide citation for the climate classification system and a brief justification for your choice.

“The most widely used climate classification system.” - Bailey, Robert G. “Ecosystem Geography: From Ecoregions to Sites” Springer, New York 2009, p.65

Citations included:

1. Köppen, W. Versuch einer Klassifikation der Klimate, vorzugsweise nach ihren Beziehungen zur Pflanzenwelt. Geogr. Z. 1900, 6, 593-611. 

2. Kottek, M.; Grieser, J.; Beck, C.; Rudolf, B.; Rubel, F. World map of the Köppen-Geiger climate classification updated. Meteorol. Z. 2006, 15, 256-263. 

Will study selection be done in duplicate with a system for breaking ties?

Yes. We have added a statement at line 161 to this effect.

“Data points will be compared and discrepancies resolved between data extractors or a third party reviewer.”

The introduction section is lengthy and can be summarized.

The discussion on outcomes following sGAS infection in the introduction has been reduced and can be appreciated in the newly submitted paper.

Study years – please update to end of 2020 at the latest.

The protocol years have been updated to reflect papers up to the end of 2020.

Other:

Report also according to MOOSE guidelines

The protocol for the systematic review has been changed to also meet the criteria set by the MOOSE guidelines. A copy of the MOOSE guideline with documentation of page and line numbers of where these criteria are met has also been uploaded. The systematic review, when produced, will adhere to both PRISMA and MOOSE guidelines. 

Exclusion criteria – consider excluding studies reporting less than 20 participants

A very good point. We did account for sample size in a non-specific way within the Joanna Briggs Institute critical appraisal checklist, but I agree that a paper with a sample size less than 20 will not be relevant to this review, therefore have added this point to the exclusion criteria.

---

## [Editor Report · Decision Letter 1]

26 Jul 2021

Protocol for the systematic review of the epidemiology of superficial Streptococcal A infections (skin and throat) in Australia.

PONE-D-20-37209R1

Dear Dr. Wiegele,

We’re pleased to inform you that your manuscript has been judged scientifically suitable for publication and will be formally accepted for publication once it meets all outstanding technical requirements.

Kind regards,

Dafna Yahav

Academic Editor

PLOS ONE
---

## [Editor Report · Acceptance letter]

30 Jul 2021

PONE-D-20-37209R1 

Protocol for the systematic review of the epidemiology of superficial Streptococcal A infections (skin and throat) in Australia. 

Dear Dr. Wiegele:

I'm pleased to inform you that your manuscript has been deemed suitable for publication in PLOS ONE. Congratulations! Your manuscript is now with our production department. 

Kind regards, 

on behalf of

Dr. Dafna Yahav 

Academic Editor

PLOS ONE